# Immobilization and docking studies of Carlsberg subtilisin for application in poultry industry

Anum Munir Rana[1]⊚, Bart Devreese[2]‡, Stijn De Waele[2]‡, Asma Rabbani Sodhozai[1]⊚, Maryam Rozi[3], Sajid Rashid[3], Abdul Hameed[1]†, Naeem Ali[1]*

1 Department of Microbiology, Biological Sciences, Quaid-i-Azam University, Islamabad, Pakistan,
2 Laboratory of Microbiology–Protein Research Unit, Ghent University, Karel Lodewijk Ghent, Belgium,
3 National Center for Bioinformatics, Quaid-i-Azam University, Islamabad, Pakistan

⊚ These authors contributed equally to this work.
† Deceased.
‡ These authors also contributed equally to this work.
* naeemali95@gmail.com

**Data Availability Statement:** All relevant data are within the paper and its supporting information files.

## Abstract

Carlsberg subtilisin from *Bacillus licheniformis* PB1 was investigated as a potential feed supplement, through immobilizing on bentonite for improving the growth rate of broilers. Initially, the pre-optimized and partially-purified protease was extracted and characterized using SDS-PAGE with MW 27.0 KDa. The MALDI-TOF-MS/MS spectrum confirmed a tryptic peptide peak with m/z 1108.496 referring to the Carlsberg subtilisin as a protein-digesting enzyme with alkaline nature. The highest free enzyme activity (30 U/mg) was observed at 50˚C, 1 M potassium phosphate, and pH 8.0. the enhanced stability was observed when the enzyme was adsorbed to an inert solid support with 86.39 ± 4.36% activity retention under 20 optimized conditions. Additionally, the dried immobilized enzyme exhibited only a 5% activity loss after two-week storage at room temperature. Structural modeling (Docking) revealed that hydrophobic interactions between bentonite and amino acids surrounding the catalytic triad keep the enzyme structure intact upon drying at RT. The prominent hygroscopic nature of bentonite facilitated protein structure retention upon drying. During a 46-days study, supplementation of boilers' feed with the subtilisin–bentonite complex promoted significant weight gain i.e. 15.03% in contrast to positive control (p = 0.001).

## Introduction

Proteases (EC 3.4) have versatile applications in various fields and are active from acidic, neutral to alkaline pH herewith possessing maximal industrial interest. The proteases own a proven significant role in wastewater treatment, pharmaceutical, food, detergent, leather, poultry, and textile industry [1]. The demand for protease use in poultry has markedly increased, promising improved body weight and health status [2]. With ever-increasing demand, protease storage and its activity under room temperature needs to be improved. So far, several

**Funding:** The authors received no special funding for this work.

**Competing interests:** the authors have declared that no competing interests exist.

techniques have opted for the storage of enzymes such as freeze-drying, spray drying, and lyophilization. Likewise, immobilization of proteins on support material is one of the most important and efficacious methods to store proteins (at room temperature) with reusability [3]. Studies have shown that an extended shelf life and enzyme stability under storage conditions is vital to retain the efficacy of enzymes [4]. Up till now, several enzymes such as protease, laccase [5], peroxidase, catalase, pectinesterase have been successfully adsorbed on a variety of solid support materials [6–10]. Carlsberg subtilisin is a highly potent class of protease with noticeable stability at extreme temperature and pH. The immobilization of carlsberg subtilisin to bentonite has not been explored in past with an aim of poultry feed supplementation. The immobilization is classfied into two types on the basis of interaction among the enzyme and support material such as physical and chemical method. The monocovelent and weaker interactions among the enzyme and support material is known as physical immobilization including the hydrophobic interactions, van der waals forces, affinity binding, ionic binding, and hydrogen bonding. Moreover, thre are four principle techniques to immobilize the enzyme to solid support material namely, entrapment, adsorption, covelant bonding and cross linking. Organic supports are generally poor with a humid and temperature environment as the shelf life is greatly reduced increasing the need of preservative addition is required to avoid microbial growth. Moreover, the inorganic supports are less intervening with the feed and enzyme to be fed eventually increasing the potential for being used in poultry as feed supplement.

One of the crucial requirements for enzyme immobilization is the selection of appropriate solid support material. The support material, either organic or inorganic by nature, is selected based on limitations, usage method, and desired application [11]. In some studies, organic supports such as carbon SKN-2P, lignocellulosic substrates, graphene oxide particles [12] and lignin have been used as solid support for the immobilization of protease and cellulase, respectively [13,14]. Ideally a support that is inert can work best for use in poultry due to its nonreactive ability, since the shelf life and environmental factors can contaminate the supports and require addition of preapproved preservative for poultry feed. Besides organic, inorganic supports serve as a good option for protein immobilization [15]. Among inorganic supports clay, due to its diverse characteristics (surface chemistry, particle size, surface area, and particle shape) is extremely suitable for enzyme immobilization. The hydrated sodium calcium alumino-silicate, zeolites, activated charcoal, and bentonite have been widely used for detoxification of feed [16]. Bentonite is a low-cost inorganic matrix that lack toxicity and chemical reactivity, making it suitable most solid support for enzyme immobilization. Since the bentonite carries high swelling capacity it is being used as sorbent to reduces wet droppings, studies reported that it also contributes toward the increase chickens' body weight, life expectancy, and egg size [17]. Additionally, an absorbed Carlsberg subtilisin to bentonite has never been exploited for its potential use in poultry as a feed supplement.

The protein-ligand docking gives an insight into possible binding regions among protein and support material that are the major contributors of interactions. Protein and ligand docking can be used as the source to predict patterns of binding proteins to specific solid support material after immobilization and also gives insight to predict reasons of attachment [18]. The docking studies is a newer approach that help find out the clues to possible stability retention after drying and poultry experiments confirms the active participation of enzyme in digesting proteins while being in the gut.

The current study was designed to improve the dietary protein assimilation among broilers. Carlsberg subtilisin isolated from *Bacillus licheniformis* PB1 was identified, characterized, and optimized for immobilization on the suitable solid support. To achieve maximum activity after adsorption, the enzyme immobilization was performed by using optimized parameters (OVAAT) viz., selection of the suitable solid support, molarity, temperature, and buffer pH

was studied. To predict the possible reason for enzyme stability under drying conditions, molecular docking interactions of two subject proteins (serine protease (3QTL) and Carlsberg subtilisin (PDB ID:4GI3) having a high resemblance to the enzyme under study) with Bentonite were studied. Furthermore, the immobilized product was fed to the Broilers (*Arbor acres*) to prospect the impact of dried-enzyme-bentonite product on the overall weight improvement of birds and compared with the commercially available protease supplement.

## Materials and methods

The work was approved by Bio-Ethical Committee (BEC-FBS-QAU2021-313) of Department of Microbiology, Quaid-i-Azam University, Islamabad.

### Chemicals

The chemicals and reagents used in this research work are of analytical grade. Chemicals such as NaCl, yeast extract, casein, peptone, gelatin, ammonium sulfate, Magnesium sulphate ($MgSO_4$), Calcium chloride dehydrate ($CaCl_2$ $2H_2O$, Dipotassium hydrogen phosphate ($K_2HPO_4$), azocasein, $\alpha$-hydroxycinnaminic acid, acetonitrile (ACN), ammonium citrate, Trifluoroacetic Acid (TFA), and KCl were purchased from Sigma-Aldrich (Merck Group, Germany). Ultracell centrifugal filter tubes purchased from Millex Amicon Ultra filter (Merck-Millipore, Darmstadt, Germany). For the SDS-PAGE protein standard Precision Plus Protein All Blue Standards was purchased from BioRad (California, United States). Woogene B & G enzyme supplement (South Korea), All reagents have been prepared in sterilized Milli Q water.

### Protease production, purification, and extraction

**Bacterial cultivation.** The alkaline protease was produced by *Bacillus licheniformis* PB1, previously isolated from desert soil of Sindh, Pakistan [19]. The stock culture was maintained on nutrient agar plates and stored in 30% glycerol at $-80°C$. The culture was grown in inoculum medium (w/v) composed of NaCl 0.5%, yeast extract 0.5%, casein 0.5%, peptone 0.3%, gelatin 0.3% with pH 8.5 and incubated at $60°C$, 160 rpm for 24 hours. The culture was used to inoculate production flasks at optimal growth inoculum ($69x10^5$ CFU/ml).

**Enzyme production.** The alkaline production medium (w/v) was prepared by adding soybean meal 1%, NaCl 0.01%, $MgSO_4$ 0.05%, $CaCl_2$. $2H_2O$ 0.05%, $K_2HPO_4$ 0.02% and KCl 0.2% with pH maintained at 8.0. This production medium (3 L) was inoculated with 2% (v/v) *B. licheniformis* and incubated at $60°C$, 160 rpm for 7 days. Later, the extracellular enzyme was separated from the cell content by centrifugation at 10,000 rpm for 20 mins at $4°C$.

**Enzyme extraction and purification.** Ammonium sulphate (w/v) at 80% saturation was added to the supernatant medium protein precipitation. Further in next line precipitated enzyme pellets were suspended in 0.05 M Tris-HCl buffer, pH 7.5. The precipitated enzyme was dialyzed using vivaspin utlraspin centrifugal dialysis tubes with cut off 10 kDa to remove excessive salts and assayed for protein content and enzyme activity.

**Protease activity assay.** The protease activity was determined by an azocasein (Sigma) assay at 440 nm [20]. Negative (without protease) and positive controls were run along to find out the activity. The total protein was estimated by Lowry assay using bovine serum albumin as a standard protein at 280 nm [21]. The enzyme activity, expressed in units, was calculated by Goose's Formula and specific activity was determined [22]. One unit of enzyme activity is defined as amount of enzyme required to liberate 1 ug of tyrosine /ml at $50°C$ and pH 8.

## Electrophoretic analysis

The enzyme was concentrated by ultrafiltration using an Ultracell centrifugal filter with a molecular weight cut-off of 10 kDa. Protein size was determined using SDS-PAGE on a 12% polyacrylamide resolving gel as reported by Laemmli et al [23]. Protein bands were observed by staining the gel with Coomassie brilliant blue G-250 and apparent molecular weight was estimated with Precision Plus Protein All Blue Standards. The 1% casein (copolymerized with 12% resolving gel) zymography was carried out according to the modified method of Garcia-Carreno [24].

## Identification of proteins by MALDI-TOF/TOF mass spectrometry

Selected protein bands were excised from SDS-PAGE followed by trypsin digestion (0.1 ug/ul) [25]. These digests were subjected to matrix-assisted laser desorption ionization-time of flight mass spectrometry (MALDI-TOF-MS). Measurements were carried out on a 4800+ MALDI TOF/TOF Proteomics Analyzer (AB-SCIEX, Darmstadt, Germany). An equal amount (1:1) of tryptic digest mixtures from each band and matrix solution (5 mg/mL α-hydroxycinnaminic acid dissolved in 60% ACN, 10 mM ammonium citrate, 0.1% TFA) was spotted on a stainless-steel MALDI target plate. The MS spectra were obtained in positive ion mode. Spectra were acquired from 5000 laser shots with a 200 Hz laser (Nd: YAG laser; 355 nm), the laser intensity was 4200 reflector measurement. The proteins were identified using the Mascot (Matrix Science) search engine against the SwissProt database (2017 version). Database search was conducted with the following parameters: three missed cleavages, carbamidomethylation of Cys, and oxidation of methionine as fixed and variable amino acid modifications, respectively, allowing an MS/MS error of 0.6 Da to fragment ions and maximum accuracy of 200 ppm for parent ions were used to identify the specific protease produced from *Bacillus licheniformis* PB1. To identify a protein, the confidence threshold was adjusted to 95%.

## Enzyme immobilization

**Free enzyme optimization.** To determine the physical parameters required for adsorption of protease to a solid support, we first assessed the effect of pH, temperature, and buffer molarity on enzyme activity. The varying molarity of potassium phosphate buffers (0.1 to 2.2 M with an interval of 0.1 M) was used to study the effect of electrolytes on enzyme activity. Likewise, a range of pH (2 to 9 with an interval of +1) was used to study its effect on enzyme activity. Furthermore, enzyme activity was determined at different temperatures (20 to 70˚C with an interval of 10˚C).

**Enzyme adsorption on selected solid supports.** The Subtilisin was adsorbed to a set of organic and inorganic support materials. The inorganic support was inert material, i.e. an active component of clay, bentonite (provided by SJ Enterprises, Pakistan). The enzyme was immobilized on bentonite by simple adsorption. Therefore, bentonite was pretreated with 1% HCl, washed several times, and dried overnight in a hot air oven at 100˚C. The dried bentonite (1 gm) was ground and equilibrated with potassium phosphate buffer (1 M, pH 8). Next, it was incubated with 2 ml of protease solution containing 15.32 U/mg (SD ± 2.5%) in the same buffer at 4˚C for 12 h at 50 rpm. Control experiments were performed without subtilisin. The unbound enzyme was removed by washing with the same buffer. Enzyme adsorbed product was further quantified to determine its specific activity. The activity yield was determined by dividing the total activity of immobilized enzyme with the total activity of free enzyme.

Wheat bran and soybean meal were selected as carbon-based solid support material. Further, optimized enzyme buffer suspension (18.34 U/mg) was adsorbed on soybean meal and wheat bran in 1:1 proportion, thin layers were spread out with the help of a spatula on Petri

plates and left for 12 hours to dry at room temperature (25 ± 3˚C) via air drying. Later, dried material was suspended in 0.05 M Tris-HCl (pH 7.5) and specific activity of protease was determined after 14 days of storage at room temperature.

**Enzyme desorption.** The total enzyme release was measured by suspending the Subtilisin bentonite dried product in Tris-HCl buffer with pH 7.5 kept at 37˚C for 10 mins under shaking conditions. After incubation, the product was centrifuged (6000 rpm for 10 mins) and supernatant was assayed to study the activity of the released enzyme in each wash was evaluated. The process was repeated four times and total adsorbed protein activity was calculated.

## Structural modeling

The X-ray crystal structure of Serine protease (3QTL) and Subtilisin (PDB ID: 4GI3) from *Bacillus licheniformis* was retrieved from Protein Data Bank [26]. The 2D structure (SDF format) of Bentonite (PubChem ID: 72941614), (Fig 4) was obtained via PubChem database, and Nanotube was constructed by combining four monomers (with a diameter of 9nm) subunits of sodium bentonite in Avogadro [27]. PDB format was generated through PyMol. The 3D structure of the nanotube was then subjected to energy minimization using GAFF force field [28].

Dockings of solid support material and its nanotube with Bacterial Serine Protease (BSP) and Subtilisin (PDB ID:4GI3) was accomplished using PatchDock server [29]. The refinement and rescoring tool FireDock was used to check the specificity of the respective interacting protein [30]. The input parameters in PatchDock were coordinate files of Serine Protease and sodium bentonite. Each enzyme transformation was calculated by scoring function to observe the geometric fit and atomic desolvation energy [31]. Root Mean Square Deviation (RMSD) clustering was engaged to the candidate solution to exclude the redundant one. The best scoring complex was subjected to UCSF Chimera ver. 1.11.2 and LigPlot for interaction analysis [32,33].

## Supplementation of Carlsberg subtilisin-bentonite product to poultry as a growth enhancer

Aiming to evaluate the use of adsorbed-dried-product on weight gain, a one-day-old commercial broiler named *Arbor acres* (n = 150) was selected and raised on a poultry farm (near Attock). The work was approved by Bio-Ethical Committee (BEC-FBS-QAU2021-313) of Department of Microbiology, Quaid-i-Azam University, Islamabad. The birds were divided into three groups consisting of 1 day-old mixed-sex broiler chickens (Cobbs), 50 birds in each. During the first 3 days, chickens were fed on a diet without enzyme supplementation. From the day onwards, for 42 days their feed was supplemented differently. Group A feed was supplemented with adsorbed protease product with the activity of 27.04 U/mg (SD ± 0.0035) on 1 gm of bentonite added to 1 kg bag of basal feed. Group B was supplemented with B & G commercially available enzyme product as a positive control 1 gm/Kg, as advised by the manufacturer's feeding chart. Group C was not supplemented with enzyme product and was run as a negative control. Both positive and negative controls have bentonite in them since it acts as a sorbent of excessive water and reduces wet litter from bird's feces. Birds were weighed at a regular interval of 6 days from 4[th] to 46[th] day.

## Statistical analysis

The experimental optimization results of three parallel measurements were assigned mean value and analysis was conducted. The results were statistically analyzed by ANOVA and Duncan's test using SPSS version 2017 software, Chicago, SPSS Inc.

## Results and discussion

### Protease production, purification, and extraction

The protease was extracted from *B. licheniformis* PB1 under batch mode agitated condition (2% inoculum, 8.5 pH, and 160 rpm at 60°C). The highest enzyme activity (231.1 U/ml) was observed on the 7th day of production [34]. Likewise, proteases were produced by *B. licheniformis* PB1 [35] under varying physiochemical production parameters. These parameters were key determinant of varying production of protease in quantity (activity) from different species. After partial purification through ammonium sulphate precipitation (80%) an about 8.21-fold increase in enzyme activity was observed with an overall yield of 59% from *B. licheniformis* PB1.

### SDS PAGE, zymography and MALDI TOF-MS/MS

SDS PAGE gel analysis of partially purified *B. licheniformis* PB1 protease revealed five bands. The band activity of Carlsberg subtilisin has been observed through zymogram analysis and a clear casein hydrolysis zone confirmed proteolytic characteristics of the enzyme (Fig 1A & 1B, S1 Fig). The protein bands from SDS-PAGE were extracted from the gel, digested with trypsin, and analyzed through MALDI-TOF MS/MS. The MASCOT database search revealed that the protein band had an apparent molecular weight of 27 kDa and is most likely a Carlsberg-subtilisin from *B. licheniformis* as reported by Pannkuk [36]. The peptide confirmation was obtained by MS/MS analysis of a tryptic peptide peak with m/z 1108.496. MS/MS showed that this peptide is identical to the peptide covering amino acid residue 342–351 (KHPNLSASQVRN) of the *apr* gene product (Mascot score 80) (Fig 1C). The Unipept analysis revealed that peptide was hitherto solely found in *B. licheniformis* Subtilisin [37]. Further the peptide sequence was found to have 40 score on NCBI protein Blast as Carlsberg subtilisin of *Bacillus licheniformis*.

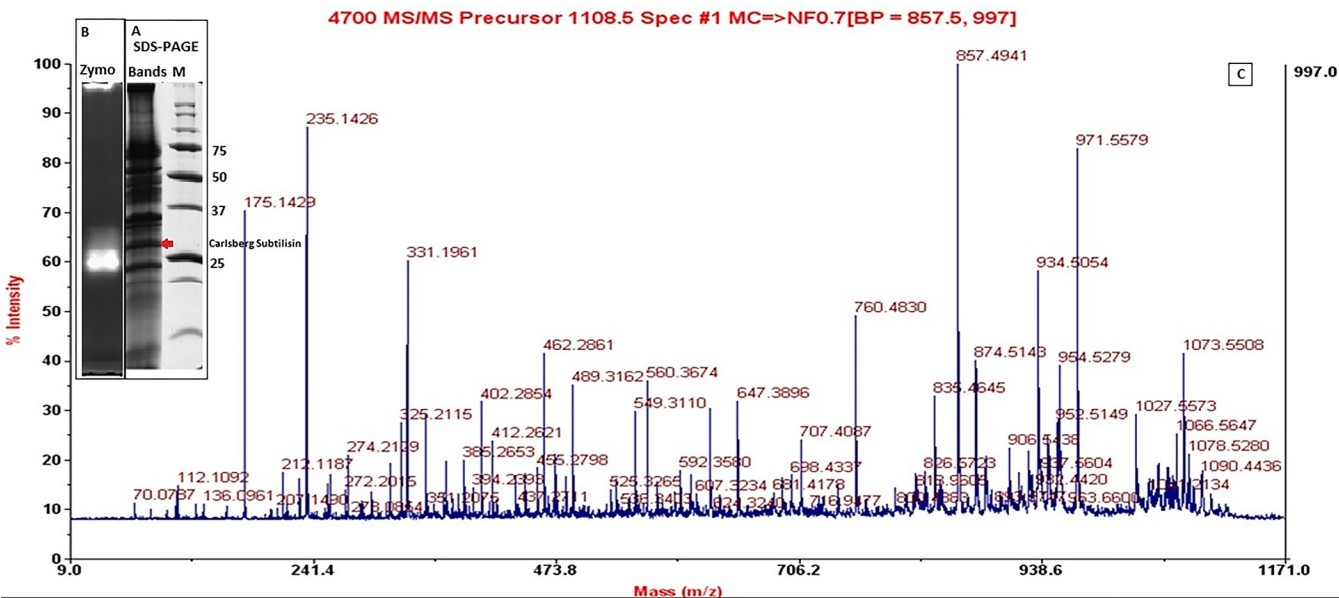

**Fig 1.** Carlsberg Subtilisin produced by *B. licheniformis* PB1 (A) SDS-PAGE Excised Gel bands with protein marker for 80% precipitated enzyme extract (B) Excised zymogram (1% Casein) of potentially purified enzyme (C) MALDI -TOF MS/MS spectrum of a tryptic peptide with M/z 1108.5 (S1 Fig). *A The SDS-PAGE was excised to visualize the thickest bands. *B The excised Zymogram of purified Carlsberg subtilisin.

Further the peptide sequence was found to have 40 score on NCBI Protein Blast as Carlsberg subtilisin of *Bacillus licheniformis*.

>sp|P00780|SUBT_BACLI Subtilisin Carlsberg OS = *Bacillus licheniformis* OX = 1402 GN = apr PE = 1 SV = 1

MMRKKSFWLGMLTAFMLVFTMAFSDSASAAQPAKNVEKDYIVGFKSGVKTASVKK DIIKE

SGGKVDKQFRIINAAKAKLDKEALKEVKNDPDVAYVEEDHVAHALAQTVPYGIPLI KADK

VQAQGFKGANVKVAVLDTGIQASHPDLNVVGGASFVAGEAYNTDGNGHGTHVAG TVAALD

NTTGVLGVAPSVSLYAVKVLNSSGSGTYSGIVSGIEWATTNGMDVINMSLGGPSGS TAMK

QAVDNAYARGVVVVAAAGNSGSSGNTNTIGYPAKYDSVIAVGAVDSNSNRASFSSV GAEL

EVMAPGAGVYSTYPTSTYATLNGTSMASPHVAGAAALILSKHPNLSASQVRNRLS STATY

LGSSFYYGKGLINVEAAAQ

**Solid support selection for enzyme adsorption.** To select appropriate solid support for adsorption of Carlsberg subtilisin, different organic and inorganic solid supports were tested for better activity after adsorption. Carlsberg subtilisin was successfully immobilized to both types of supports but overall activity was different. The adsorption on carbon-based solid support viz., soybean meal exhibited 27% greater activity (0.33 ± 0.007 U/mg) than that of wheat bran (0.25 ± 0.0014 U/mg). Moreover, bentonite immobilized Carlsberg subtilisin (0.45 ± 0.0071 U/mg) retained better residual activity in contrast to carbon-based solid supports (98%) (Fig 2A). This indicated bentonite has better immobilization ability as compared to carbon-based support material and the overall activity of enzyme was improved markedly. Similar results were previously obtained for α-amylase bound to chitosan and Na-bentonite composites with an overall increase in the activity (87%) of enzyme [38]. For prolonged storage of proteins, carbon-based immobilization support materials are not advisable for regions that have a wider range of temperature and humidity, increases the chance of fungal contamination. The storage capacity on wheat bran and soybean meal as solid support appeared to be very low and changing environmental conditions may contaminate the product and lead towards harmful aflatoxins production by fungal strain [39]. As the poultry is a very sensitive organism and mild fungal contamination in feed can possibly lead to loss of organism.

## Optimization of conditions before adsorption

**Effect of change in buffer molarity.** The overall enhanced activity was observed with an increase in molarity of buffer until the optimum (1M potassium phosphate, pH 8) and it was dropped on further increase. Inclusion of potassium phosphate buffer (1 M) leads to 10-fold enhanced specific activity 14.805 ± 0.57 U/mg with respect to the crude enzyme, elucidating the marked impact of electrolyte strength on enzyme activity (Fig 2B). Likewise, McComb *et al*., revealed activity also relied on electrolyte molarity under consideration [40]. Therefore, the results strongly support that Carlsberg subtilisin exhibited a stable activity around higher electrolyte strength.

**Free enzyme temperature optimization.** In general, the temperature has a remarkable impact on Subtilisin activity. The specific activity of Subtilisin was found to be 33.667 ± 0.94 U/mg at 50 ˚C (Fig 2C). The enzyme is still active at a range of higher temperature up to 70˚C after 30 min of incubation. Our findings coincide with a study reporting 50 ˚C as optimal

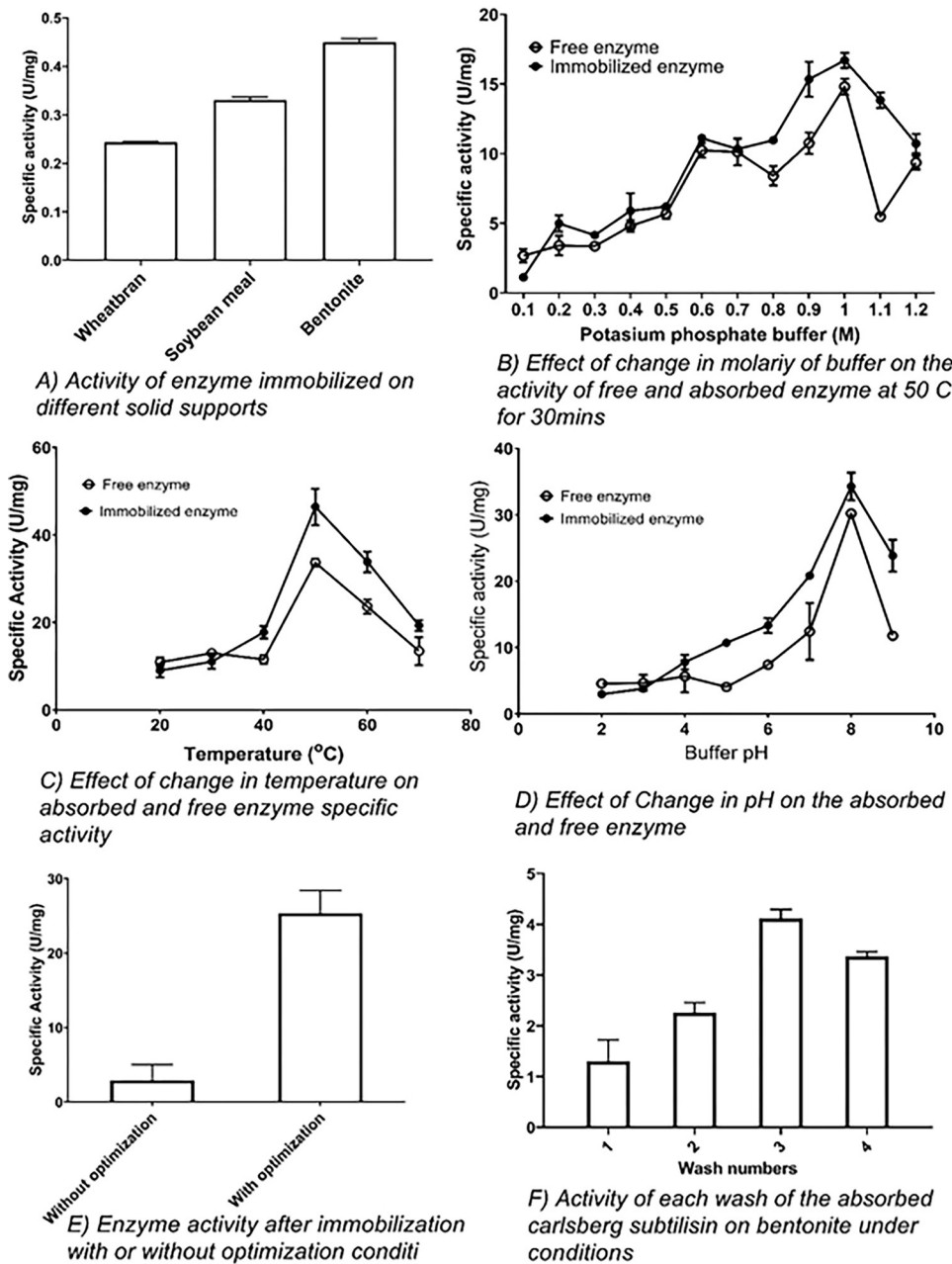

**Fig 2. Free and absorbed enzyme optimization for immobilization to a suitable solid support and desorption studies.**

temperature for Subtilisin. Although, the latter enzyme had a wider temperature range (30–60 ˚C) for its pronounced activity [41]. In terms of thermal stability, the activity of Subtilisin extracted from *B. licheniformis* PB1 was better as compared to other Bacillus species such as *B. subtilis* [42].

**Influence of pH for enzyme activity.** A wide range of Subtilisin has been produced from different *Bacillus* species with a relatively high optimal pH [43]. The specific activity of Subtilisin at optimum pH (8.0) was recorded $30.2 \pm 0.26$ U/mg indicating enhanced activity as compared to crude enzyme due to protein enrichment (Fig 2D). This specific activity could be due

to an overall charge at the interfacial enzymatic surface. This charge distribution on the enzyme surface facilitates substrate binding and catalytic efficiency, increasing overall enzyme-substrate complex formation. In other words, there is an optimum pH value that favors the maximum concentration of the enzyme-substrate intermediate [44].

**Free enzyme adsorption on a selected solid support.** Successful immobilization of Carlsberg-Subtilisin could be a consequence of adsorption through hydrophobic interactions among enzymes and supporting material [45]. It can be said that involvement of possible strong interactions among enzymatic amino acids (essential for bonding or catalysis) and the surface of the solid support (clay mineral) are associated with successful immobilization [46]. This could affect the structure of active center surroundings and subsequently the final catalytic features of lipase [3].

**Enzyme desorption.** The extent of enzyme desorption from adsorbed carriers could be evaluated by suspending the product in the buffer. The Subtilisin adsorbed to bentonite in dried form was suspended in 1 M Potassium phosphate buffer pH 8 and placed for 10 mins at room temperature, later centrifuged at 6000 rpm. The overall desorption of Subtilisin from the product was 28.89 ± 3.24% in 40 mins after four washes. Consequently, the protein desorption was achieved with approximately 71.1 ± 4.1% of active protein attached to the solid support after concomitant four washes (Fig 2F).

**Molecular docking analysis.** Despite the difference, in structure, both enzymes were evaluated to validate the adsorption of the protease. Molecular docking analysis of Serine protease (3QTL) and Carlsberg-Subtilisin (PDB ID:4GI3) against bentonite was accomplished. To further characterize the serine protease and bentonite interaction, we mapped the bentonite-specific probable regions required for Serine protease and Carlsberg-Subtilisin binding. The 10 best docking solutions were designated for additional enhancement and rescoring scrutiny by the FireDock algorithm. The binding energy of the serine protease and bentonite complex was -73.94 kcal/mol, while the binding energy of Carlsberg subtilisin and bentonite was -47.58 kcal/mol. In both Serine protease-sodium bentonite and Carlsberg subtilisin-bentonite complexes, catalytic triad (Ser220, His63, and Asp32) and (Ser221, His64, and Asp32) residues were involved in the formation of substrate-binding clefts, respectively (Fig 3A and 3B).

Positioning of bentonite onto the surface of a serine protease and Subtilisin was keenly monitored to explore the binding pocket dynamics and residual contributions of protein in

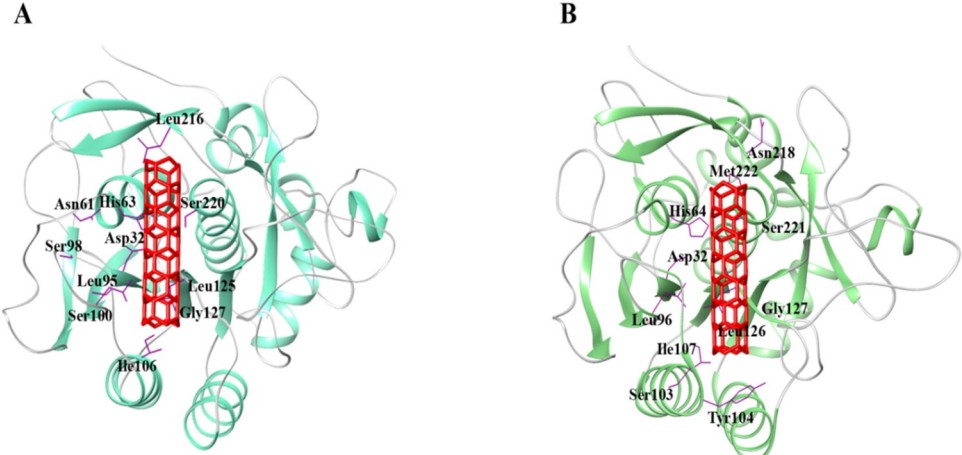

**Fig 3.** Detailed interaction of best-docked complexes with (A) Serine protease (3QTL) (B) Subtilisin (PDB ID: 4GI3). Protein is styled in ribbon representation while interacting residues are depicted in wire form with labeled residues in black color. Bentonite nanotube is illustrated in stick representation with red-colored (using UCSF Chimera).

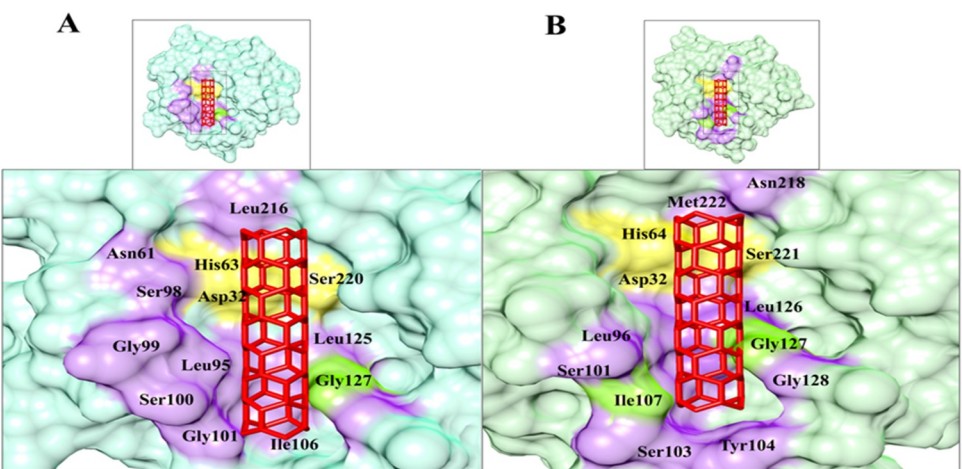

**Fig 4. The best-docked pose of bentonite nanotube in the BSP binding cleft.** Surface view of (A) Serine protease (3QTL) (cyan) and (B) Subtilisin (PDB ID: 4GI3) -specific bentonite binding cavity. The binding pocket is indicated by a light purple colored surface onto Subtilisin (light green) while interacting residues are labeled in black. Catalytic binding pocket residues in Subtilisin are indicated by yellow color and the residue implicated in the hydrogen bonding is indicated by light green color. The bentonite nanotube is indicated by the red colored stick (UCSF Chimera).

association with docked ligand (Fig 4A and 4B). revealed the predominant contributions of hydrophobic residues lying in the periphery of active sites. Noticeably, His63, Leu95, Gly99, Ser100, Gly101, Ser102, Tyr103, Ile106, Gly126, Gly127, Asn154, Leu216, and Asn217 residues of serine protease were involved in hydrophobic interactions with sodium bentonite. In contrast, Subtilisin-specific His64, Leu96, Gly100, Ser101, Gly102, Ser103, Tyr104, Ile107, Ser125, Leu126, Gly128, Asn155, Leu217 and Asn218, Ser221, Met222 residues were involved in bentonite binding (Fig 5A and 5B). In the Serine Protease-bentonite complex, Gly127 residue was involved in hydrogen bonding. Moreover, Ile107 and Gly127 residues of Subtilisin were also found in hydrogen bonding.

The docking analysis revealed significant involvement of van der Waals, Water, and desolvation energies in the binding of bentonite to Ile107, Gly127 residues of subtilisin exhibited more conformational changes as compared to Serine Protease due to binding. Likewise, Gilli *et al.*, reported that acidic or basic medium plays a vital role in the establishment of a strong hydrogen bond or strengthens it to 6-folds followed by a 15% shorter bond [47]. This could be the reason for the establishment of stronger hydrogen bonds among bentonite and protease keeping the catalytic triad intact. Despite drying protease and bentonite, enzyme activity was sustained, and only a limited amount of activity was lost (how much). Additionally, the pH of Cobb's gut varies greatly from acidic to mild basic (pH 3–7.5) that supported meliorate in desorption rate [48]. This could be a reason for increased weight gain when the dried product was subjected to poultry trials.

## Poultry application

The Carlsberg-subtilisin-bentonite product prepared as a result of immobilization, subjected to poultry feed with aim of finding impact on chicken weight, when fed in comparison to a commercially available product. A gradual increasing trend was observed among treatment groups (A, B, and C) to the total number of days, product was fed to chickens. A significant weight gain was observed on day 10, 28, 34, 40, and 46. Therefore, treatment group A has a significant difference in weight gain as compared to group B (positive control) and C (negative control) showed in Table 1. This is in line with a previous study that proved visibly increased

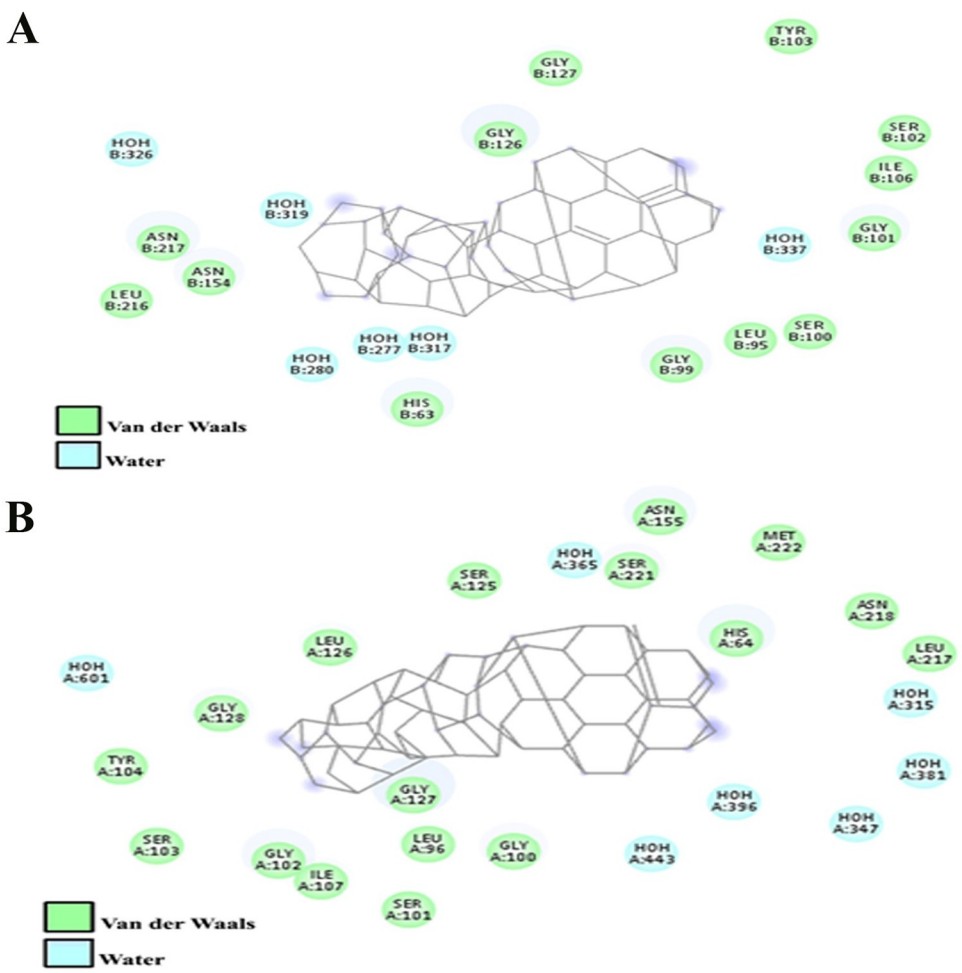

**Fig 5.** Inter-molecular interaction pattern of the **(A)** Serine protease (3QTL) and (B) Subtilisin (PDB ID:4GI3) to Sodium bentonite. 2D structure of bentonite is indicated in the middle while interacting residues are illustrated by green balls. The water molecules are indicated by cyan-colored balls.

**Table 1. Comparison of enzyme immobilization to organic and inorganic solid supports.**

| Support Material | Enzyme | Immobilization Technique | Percent immobilization | Ref. |
|---|---|---|---|---|
| **Inorganic Support** | | | | |
| bentonite-derived mesoporous materials | Laccase | Physical adsorption | 60% removal efficiency | [51] |
| Na-bentonite and modified bentonite | Alkaline phosphatase | Cation exchange and adsorption by van-der-Waals interaction | | [46] |
| Bentonite | lipase | | (30%) | [52] |
| Na-bentonite and modified bentonite | α-amylase | Adsorption | | [53] |
| Organobentonite | Lipase | Adsorption | $BC_{100}$-lipase kept 82.5% of initial activity | [54] |
| **Organic Support** | | | | |
| Nanoporous rice husk silica | Soybean lipoxygenase | Adsorption | 50% | [55] |
| DEAE-cellulose | α-amylase | Adsorption | 80% | [56] |

**Table 2. Weight gain in Broilers (46 Days) after addition to feed Carlsberg-subtilisin-bentonite complex, B&G enzyme product, and without enzyme.**

| | Treatment groups | | | | |
|---|---|---|---|---|---|
| Age (days) | A | B | C | SDM* | SEM** |
| 4 | 46.86[b] | 47.2[a] | 47.3[a] | 1.09 | .08980 |
| 10 | 168.06[a] | 151.64[a,b] | 155.22[a,b] | 12.825 | 1.05066 |
| 16 | 343.92[a] | 333.54[b] | 318.42[c] | 22.21 | 3.14040 |
| 22 | 522.9[a] | 466.77[c] | 449.14[c] | 26.96 | 3.81308 |
| 28 | 863.8[a] | 754.54[b,c] | 735.92[b,c] | 41.98 | 7.61214 |
| 34 | 1280.2[a] | 1113.44[b,c] | 1072.73[b,c] | 62.9 | 16.30792 |
| 40 | 1613.2[a] | 1387.46[b] | 1351.5833[b] | 84.84 | 26.67894 |
| 46 | 1821.8[a] | 1583.76[b] | 1554.3958[b] | 86.95 | 26.98344 |

[a-c] Means in a row lacking common superscript letters differ significantly (P <0.05), Duncan's test.

Note

Treatment group A: Protease product.

Treatment group B: Phytezyme (commercial product as positive control).

Treatment group C: Control (no enzyme product).

*Standard Deviation Mean.

**Standard Error Mean.

body weight of chicken vs. time (days) when fed on an enzyme-based diet. In contrast to 129 gm weight on the 7[th] day, our results explicitly showed an efficient increase in weight gain from day 4 to day 10[th] (168.06 gm) [49]. Treatment group A retained significantly higher weight gain (1821.8 gm) compared to the treatment group B and C. In this experiment, the enzyme continued to demonstrate catalytic activity in terms of weight gain. This could be an advantage for broiler feeding, since enzyme supplementation typically reduces overall digesta's viscosity, which influences feed retention, lowering it from 14–11 hour (Table 2) [50]. Therefore, the addition of protease inorganic sorbent (bentonite) products slowed down the feed passage to an acceptable level. It can be inferred that approximately 11 hours of feed retention time, results in a slower release of enzymes from harnessing sorbent thus facilitating the hydrolysis of peptides throughout the gastrointestinal tract. Eventually, increased peptide assimilation was estimated in terms of prominent weight gain. The adsorbed subtilisin efficiently hydrolyzed protein-rich basal medium and enhanced peptide assimilation despite the bound state of the enzyme. Therefore, an overall ameliorated weight gain in chicken till 46 days was observed to be 15.03% as compared to the positive control. The adsorption of subtilisin to bentonite participates in securing the activity of the enzyme under drying conditions by keeping the active site intact.

## Conclusions

Protein catalyzing enzymes have diverse application in industry, specifically in poultry broilers, proteases are used as the feed supplement. For which proteases like enzyme was produced from *Bacillus licheniformis* PB1 and extracted, partially purified, dialyzed, and characterized using MALDI-TOF MS/MS to identify the specific protease involved in the immobilization is an efficient way to enhance protein stability (S1 & S2 Figs in S1 Raw images). With this property, numerous solid supports (organic and inorganic) have been reported as a promising solution to increase the shelf life of biological catalysts and contribute towards commercial utility. Where clay components' enduring hydrophobic and adsorptive characteristics can be a good choice for protease immobilization. However, immobilization of Carlsberg-subtilisin to solid

support required optimization of parameters to provide cushioning of the enzyme along with better adsorption on bentonite. Based on the results of present study, adsorbed-product contributed better stability at RT, and docking experiments gave a possible explanation. To be more specific, docking results confirmed that the presence of specific hydrophobic interactions among catalytic triad and bentonite is responsible for pronounced stability and activity of protein after adsorption. According to the present study, the active center of subtilisin is in close vicinity to the bentonite surface that created a hydrophobic environment around the active center compared to the aqueous medium surrounding dissolved enzyme.

## Supporting information

**S1 Fig. Carlsberg subtilisin produced by *B. licheniformis* PB1.** (A) SDS-PAGE Excised Gel bands with protein marker for 80% precipitated enzyme extract (B) Excised zymogram (1% Casein) of potentially purified enzyme (C) MALDI -TOF MS/MS spectrum of a tryptic peptide with M/z 1108.5.
(PDF)

**S1 Raw images.**
(PDF)

**S1 File.**
(DOCX)

## Author Contributions

**Conceptualization:** Anum Munir Rana, Abdul Hameed.

**Data curation:** Stijn De Waele, Naeem Ali.

**Formal analysis:** Anum Munir Rana, Bart Devreese, Maryam Rozi, Sajid Rashid.

**Investigation:** Anum Munir Rana, Asma Rabbani Sodhozai, Naeem Ali.

**Methodology:** Naeem Ali.

**Project administration:** Anum Munir Rana, Naeem Ali.

**Resources:** Bart Devreese, Abdul Hameed.

**Software:** Maryam Rozi.

**Supervision:** Sajid Rashid, Naeem Ali.

**Validation:** Bart Devreese, Sajid Rashid.

**Visualization:** Stijn De Waele, Sajid Rashid.

**Writing – original draft:** Anum Munir Rana, Maryam Rozi, Sajid Rashid.

**Writing – review & editing:** Bart Devreese, Stijn De Waele, Asma Rabbani Sodhozai, Naeem Ali.

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
