## [Decision Letter · Decision Letter 0]

1 Jun 2021

PONE-D-21-14945

Immobilization and docking studies of Thermophilic Carlsberg Subtilisin on Bentonite as a poultry feed supplement

PLOS ONE

Dear Dr. ali,

Thank you for submitting your manuscript to PLOS ONE. After careful consideration, we feel that it has merit but does not fully meet PLOS ONE’s publication criteria as it currently stands. Therefore, we invite you to submit a revised version of the manuscript that addresses the points raised during the review process.

We look forward to receiving your revised manuscript.

Kind regards,

Sanjay Kumar Singh Patel, Ph.D.

Academic Editor

PLOS ONE

Journal Requirements:

2. To comply with PLOS ONE submissions requirements, in your Methods section, please provide additional information on the animal research and ensure you have included details on animal welfare and efforts to alleviate suffering.

3.PLOS ONE now requires that authors provide the original uncropped and unadjusted images underlying all blot or gel results reported in a submission’s figures or Supporting Information files. This policy and the journal’s other requirements for blot/gel reporting and figure preparation are described in detail at https://journals.plos.org/plosone/s/figures#loc-blot-and-gel-reporting-requirements and https://journals.plos.org/plosone/s/figures#loc-preparing-figures-from-image-files. When you submit your revised manuscript, please ensure that your figures adhere fully to these guidelines and provide the original underlying images for all blot or gel data reported in your submission. See the following link for instructions on providing the original image data: https://journals.plos.org/plosone/s/figures#loc-original-images-for-blots-and-gels.

Reviewers' comments:

Reviewer's Responses to Questions

**Comments to the Author**

1. Is the manuscript technically sound, and do the data support the conclusions?

Reviewer #1: Yes

Reviewer #2: Yes

Reviewer #3: Partly

2. Has the statistical analysis been performed appropriately and rigorously? 

Reviewer #1: Yes

Reviewer #2: Yes

Reviewer #3: Yes

3. Have the authors made all data underlying the findings in their manuscript fully available?

Reviewer #1: Yes

Reviewer #2: Yes

Reviewer #3: Yes

4. Is the manuscript presented in an intelligible fashion and written in standard English?

Reviewer #1: Yes

Reviewer #2: Yes

Reviewer #3: Yes

5. Review Comments to the Author

Reviewer #1: This manuscript report about Immobilization and docking studies of Thermophilic Carlsberg Subtilisin on Bentonite as a poultry feed supplement. Manuscript needs following changes for further improvement.

Comments:

Introduction: Improve this section by adding more information about properties of matrix required for enzyme immobilization and various supports reported for protease immobilization.

Line 31: Add suitable reference in support: Molecular biology interventions for activity improvement and production of industrial enzymes.

Line 51: What are drawbacks of organic support used for immobilization? Why we need inorganic support. Need explanation.

Line 54: What are the advantages of clay over other inorganic supports reported?

Check the manuscript for typo errors, there should be space between value and units.

Line 165: Check and correct… pernatant was assayed

Protease was partially purified and used in experiments. There may be several other protein which have positive or negative effect on chicken growth improvement. Need justification.

What about the Native-PAGE? Is the reported protease in its monomeric form or multimeric? Add information.

Why there is no data about the effect of pH, buffer strength and temperature effect on immobilized enzyme. This information is important for comparison. Add data and discussion.

Reviewer #2: Manuscript title: Immobilization and docking studies of Thermophilic Carlsberg Subtilisin on Bentonite as a poultry feed supplement

The manuscript describes very simple method for the preparation of protease immobilized bentonite. The resultant material used for poultry feedstock to achieve healthy and productive chicken. Additionally docking study was also performed to find optimal binding between protein and ligand. Overall, the manuscript has been written well. In my opinion, the work may be accepted for publication in PLOS ONE after Major revision.

My specific comments are as follows:

1. Reduce some words in title and change the keywords which areal ready present in title

2. Line 17: The highest free enzyme activity (a maximum of 19.02-fold) was observed at 50 °C, 1M potassium phosphate, and pH 8.0. the enhanced stability was observed when 19 the enzyme was adsorbed to an inert solid support with 86.39±4.36% activity retention under 20 optimized conditions.

Mention the maximum enzyme activity in term of unit and rewrite the sentences.

3. Abstract: Line 14. Significance of partially-purified enzyme?

4. The introduction section is lack of focus. Please be selective on the background content and rewrite the introduction part after a wider literature investment.

5. Line 36 Mention how the application of protease in poultry feed is promising? Kindly highlight productivity and improved health via control on poultry zoonotic pathogens.

6. Author must mention the types of immobilization methods in brief and advantage of the method by which immobilization has been reported in this study.

Line 43 Consider the relevant works published within 5 year for possible citations such as

Suman, Sunil Kumar, Padma Lata Patnam, Sanjoy Ghosh, and Suman Lata Jain. "Chicken feather derived novel support material for immobilization of laccase and its application in oxidation of veratryl alcohol." ACS Sustainable Chemistry & Engineering 7, no. 3 (2018): 3464-3474. https://doi.org/10.1021/acssuschemeng.8b05679

Imam, Arfin, Sunil Kumar Suman, Raghuvir Singh, Bhanu Prasad Vempatapu, Anjan Ray, and Pankaj K. Kanaujia. "Application of laccase immobilized rice straw biochar for anthracene degradation." Environmental Pollution 268 (2021): 115827. https://doi.org/10.1016/j.envpol.2020.115827.

Sanjay KS PATEL, Raviteja PAGOLU, DIBYA BHOL, LEE Jung-Kul

Eco-Friendly Composite of Fe₃O₄-Reduced Graphene Oxide Particles for Efficient Enzyme Immobilization. ACS applied materials & interfaces, 9(3), 2213-2222.

Sanjay KS Patel, Rahul K Gupta, Sang-Yong Kim, In-Won Kim, Vipin C Kalia, Jung-Kul Lee, Rhus vernicifera Laccase Immobilization on Magnetic Nanoparticles to Improve Stability and Its Potential Application in Bisphenol A Degradation Indian Journal of Microbiology 61, no. 1 (2021): 45-54.

More relevant study may be cited related to immobilization support.

7. Recheck entire manuscript for additional space, for instance line 45.

8. If the protease producing strain has deposited in any microbial culture collection repository mention the ID number (section bacterial cultivation)

9. This production medium (3L) was inoculated with 2% (v/v) B. licheniformis and incubated at

99 60°C for 7 days. Author must mention other production parameter such as RPM and pH for protease production.

10. Correct the sentence, line 102, further in next line precipitated enzyme pellets were suspended in 0.05M Tris-HCl 104 buffer, pH 7.5, and assayed for protein content and enzyme activity. The precipitated enzyme was dialyzed using vivaspin utlraspin centrifugal dialysis tubes with cut off 10 kMW to remove excessive salts. Salts interfere with enzyme activity so it is necessary to remove the excess salt from the suspended buffer before taking the enzyme activity. The sentence mentioned (line 103) indicate that, enzymatic activity was assayed first then dialysis was performed. Change the sequences of experiment and rewrite the section.

11. Protease activity method and calculation should be explained in brief for readers,

Line 110 Please add space between numbers and unit, for instance 280 nm (space between 280 and nm, please make further corrections in all the text.)

12. Line 121 Digestion protocol not provided

13. Section 2.4 what was the peptide concentration for MALDI-TOF MS/MS analysis?

14. Line 130 how the database search was performed? Any specific genera or generic search kindly mention?

15. Line 132 Mention the peptide charge used in MASCOT search

16. Kindly provide peptide sequence showing identity with protease of Bacillus licheniformis along with score obtained in MASCOT search.

17. Reason for selection of clay as support material?

18. Line 148 significance of HCl pretreatment?

19. Line 168 correct as supernatant

20. Line 208 -211” Likewise, proteases were produced by B. licheniformis 209 PB1 (33) as well but under varying production parameters. Among them, physicochemical 210 parameters were a key determinant of varying production of protease in quantity and activity from 211 different species:. Rewrite the sentences for better clarity

21. Section 3.1 before mentioning the fold activity, author should mention the actual enzyme activity in terms of IU/ml

22. Section 3.1 line 207 “highest enzyme activity observed on seventh day” is the activity reduced or stable ,what was the reason for the reduction of activity explain in brief (1-2)lines.

23. Line 221“The peptide confirmation was obtained by MS/MS analysis of a tryptic peptide

peak with m/z 1108.496. MS/MS showed that this peptide is identical to the peptide covering amino acid residue 342-351 (K.HPNLSASQVR.N) of the apr gene product (Mascot score 80)” kindly justify the statement.

24. Section 3.3 line 235 Reason for better residual activity in bentonite immobilized subtilisin compared to other support used in the study?

25. Author studied the effect of change in buffer Molarity, and found at 1M strength the activity was maximum, at what temperature this activity was measured. Include this information.

26. Line 257, It has mentioned the enzyme is active at higher temperature range up to 70°C after 30 min of incubation. The loss in activity must be compared with the maximum activity at temperature 50 0 C.

27. Section 3.4.4, is not explained adequately. The adsorption of enzyme on a selected solid support should be explain properly.

28. Line 331“only a limited amount of activity was lost (how much).” check the statement and rewrite.

29. Section 3.6 I strongly suggest authors to compare the data of poultry trials with previously reported results. This may add merit to the present study and would help readers to understand the advantage of the work.

30. Section 3.5 This section should be more descriptive rather than straight forward observation of docking results. Mention about the ligand internal coordinates in grid for protein pocket calculation, score and binding energy with comparing where applicable.

31. Spell check the entire manuscript.

32. Check the units properly and follow standard units.

33. Check your reference format. It should be uniform.

Reviewer #3: Manuscript number: PONE-D-21-14945

Title: Immobilization and docking studies of Thermophilic Carlsberg Subtilisin on Bentonite

as a poultry feed supplement

The manuscript entitled ‘Immobilization and docking studies of Thermophilic Carlsberg Subtilisin on Bentonite as a poultry feed supplement' is an interesting piece of work that gives details about immobilized thermophilic Carlsberg Subtilisin on bentonite and its application to poultry feed supplement'. They prepared the Carlsberg Subtilisin enzyme, and then immobilized on to bentonite surface by physical adsorption, and then feed to poultry feeding. The results are deserved to be published in enzyme technology area. However, the level of results and discussion is not considered enough to merit its publishing in PLOS ONE. Some of the suggested changes to be made are given below.

Major

Q1. The authors should describe advantage of bentonite relevant with enzyme immobilization. This discussion would be very important to address originality of the manuscript.

Q2. Physico-chemical characterization of bentonite should be discussed.

Q3. The author conclude that bentonite is better than carbon materials for protease immobilization. However, depending on experimental condition, the result might be changed. This information must be clear in the manuscript.

Q4. Provide the supporting literature information about bentonite for enzyme immobilization (additional comparison Table) to justify the significance of this study.

Q5. Was determined the full loading of immobilized enzyme prepared under the optimized immobilization conditions? This information must be clear in the manuscript.

Q6. The loading capacity of enzyme immobilizaed prepared and the amount of attached enzyme: What were the optimum conditions? Please use numbers, no qualitative statements (high, good, higher, etc).

Q7. The kinetics and specific activity of immobilized protease on the bentonite based on the enzyme loading need to be more quantitative and should be given and discussed in the manuscript: Authors need to compare these results with other results in the literature.

Q8. Comparison of the free and immobilized enzyme in terms of kinetic parameters: Authors need to compare these results with other results in the literature.

Q9. The activity and stability of immobilized enzymes in bentonite carrier: The new biocatalysts presented were compared with a commercial material?? This information must be clear in the introduction.

Q10. The influence of metal ions and some inhibitors on the enzyme activity were examined?

Q11. The effect of the detergents and organic solvents was studied?

Q12. All discussion sections relevant with enzyme immobilization are very week, authors can improve it by suggesting the above citations such as how various in loading? why high activity? kinetic changes? And low reusability (may be leaching), what benefits of this system? etc.

Q13. Bentonite is mineral materials which might affect weight gain to chicken. However, the author didn’t include any proper experimental condition to discuss with potential synergetic effect of them. This information must be clear in the manuscript.

6. PLOS authors have the option to publish the peer review history of their article (what does this mean?). If published, this will include your full peer review and any attached files.

Reviewer #1: **Yes: **Shashi Kant Bhatia

Reviewer #2: No

Reviewer #3: No

---

## [Author Response · Author response to Decision Letter 0]

10 Jan 2022

Reviewer #1: This manuscript report about Immobilization and docking studies of Thermophilic Carlsberg Subtilisin on Bentonite as a poultry feed supplement. Manuscript needs following changes for further improvement.

Comments:

Introduction: Improve this section by adding more information about properties of matrix required for enzyme immobilization and various supports reported for protease immobilization.

Thank you for highlighting the weak ends in the manuscript. The introduction is modified and additional information is added to the manuscript as per suggestion.

Line 31: Add suitable reference in support: Molecular biology interventions for activity improvement and production of industrial enzymes.

 I will be publishing a whole study on this aspect. As it is more intricate and requires a lot of attention so a separate study is designed to address this aspect. 

Line 51: What are drawbacks of organic support used for immobilization? Why we need inorganic support. Need explanation. 

Organic supports are generally poor with a humid and temperature varying environments as the shelf life is greatly reduced and a need of addition of preservative are required to avoid microbial growth. For which an additional approval of poultry friendly preservatives is required to be tested and approved by the DRAP. – shelf life issue, microbial growth etc. 

The inorganic supports are less intervening with the feed and enzyme to be fed eventually increasing the potential for being used in poultry as feed supplement. Also the bentonite is already a frequently used product in poultry industry to control wet liter. 

Line 54: What are the advantages of clay over other inorganic supports reported?

Check the manuscript for typo errors, there should be space between value and units.

Majorly clay and its component is already being used poultry feed with an aim to control wet liter and adsorb trace amounts of aflatoxins present in feed. Additionally, the suggested spaces has been added between the values and units. 

Line 165: Check and correct… pernatant was assayed

The typo errors has been removed after carefully review of paper.

Protease was partially purified and used in experiments. There may be several other proteins which have positive or negative effect on chicken growth improvement. Need justification.

We have designed another project and currently working on the comparative studies of the highly purified respective enzyme and partially purified. The formulations are being fed to the same bird types to test th impact of variation among the reported results. 

What about the Native-PAGE? Is the reported protease in its monomeric form or multimeric? Add information.

Yes, true this is a very valid point. That experiment was not scope of the studies. We have mentioned above that we are working on the next part of this study where we are going to run comparison between purified and partially purified proteins to check if there are any other protein is participating and causing the effect. so we have included this experiment in our work flow to explore the potential participation of other proteins causing the net effect on the bird.

Why there is no data about the effect of pH, buffer strength and temperature effect on immobilized enzyme. This information is important for comparison. Add data and discussion.

Though the data was expected to be published in another manuscript, yet I have added the data in figures to highlight the successful immobilization impact. The discussion of immobilization variations and results will change the gist of the current aspects. 

Reviewer #2: Manuscript title: Immobilization and docking studies of Thermophilic Carlsberg Subtilisin on Bentonite as a poultry feed supplement

The manuscript describes very simple method for the preparation of protease immobilized bentonite. The resultant material used for poultry feedstock to achieve healthy and productive chicken. Additionally, docking study was also performed to find optimal binding between protein and ligand. Overall, the manuscript has been written well. In my opinion, the work may be accepted for publication in PLOS ONE after Major revision.

My specific comments are as follows:

1. Reduce some words in title and change the keywords which areal ready present in title

We have changed the title word count 14 to 12. And please let us know that if there is anything else you suggest in the proposed title, if something is not correct will be highly appreciated. 

2. Line 17: The highest free enzyme activity (a maximum of 19.02-fold) was observed at 50 °C, 1M potassium phosphate, and pH 8.0. the enhanced stability was observed when 19 the enzyme was adsorbed to an inert solid support with 86.39±4.36% activity retention under 20 optimized conditions.

The maximum units in U/ml are added to the manuscript as per suggestion. 

Mention the maximum enzyme activity in term of unit and rewrite the sentences.

3. Abstract: Line 14. Significance of partially-purified enzyme?

The partially purified enzymes are proved to be economically feasible and efficient for poultry field application. Because highly purified proteins are only required for the pharmaceutical application and if highly purified product will be used eventually compromise the cost effectiveness of the product. 

4. The introduction section is lack of focus. Please be selective on the background content and rewrite the introduction part after a wider literature investment.

The introduction is improved as per suggestion and necessary additions are made to the manuscript. 

5. Line 36 Mention how the application of protease in poultry feed is promising? Kindly highlight productivity 

and improved health via control on poultry zoonotic pathogens.

A separate set of experiments were designed and executed to observe the zoonotic pathogen control. The data is part of another manuscript. We collected the fecal matter of the tested birds in triplicate from all of the groups and were cultured to identify unique microflora present in the bird fecal matter. Further the microbes were identified. We observed feeding enzyme containing formulation increased the growth of lactobacillus species which was further tested on the potentially zoonotic pathogens and tested the antimicrobial efficacy against these pathogens affirming that feeding these formulations support growth of microbes that have greater antimicrobial activity against the zoonotic pathogens. 

6. Author must mention the types of immobilization methods in brief and advantage of the method by which immobilization has been reported in this study.

The types of immobilizations are mentioned in the introduction section. 

Line 43 Consider the relevant works published within 5 year for possible citations such as

Suman, Sunil Kumar, Padma Lata Patnam, Sanjoy Ghosh, and Suman Lata Jain. "Chicken feather derived novel support material for immobilization of laccase and its application in oxidation of veratryl alcohol." ACS Sustainable Chemistry & Engineering 7, no. 3 (2018): 3464-3474. https://doi.org/10.1021/acssuschemeng.8b05679

Imam, Arfin, Sunil Kumar Suman, Raghuvir Singh, Bhanu Prasad Vempatapu, Anjan Ray, and Pankaj K. Kanaujia. "Application of laccase immobilized rice straw biochar for anthracene degradation." Environmental Pollution 268 (2021): 115827. https://doi.org/10.1016/j.envpol.2020.115827.

Sanjay KS PATEL, Raviteja PAGOLU, DIBYA BHOL, LEE Jung-Kul

Eco-Friendly Composite of Fe₃O₄-Reduced Graphene Oxide Particles for Efficient Enzyme Immobilization. ACS applied materials & interfaces, 9(3), 2213-2222.

Sanjay KS Patel, Rahul K Gupta, Sang-Yong Kim, In-Won Kim, Vipin C Kalia, Jung-Kul Lee, Rhus vernicifera Laccase Immobilization on Magnetic Nanoparticles to Improve Stability and Its Potential Application in Bisphenol A Degradation Indian Journal of Microbiology 61, no. 1 (2021): 45-54.

More relevant study may be cited related to immobilization support.

The suggested recent papers on the enzyme immobilization and efficacy of enzyme has been mentioned in the manuscript to improve the introduction section. 

7. Recheck entire manuscript for additional space, for instance line 45.

The additional spaces are reviewed carefully and changes are made as per suggestions.

8. If the protease producing strain has deposited in any microbial culture collection repository mention the ID number (section bacterial cultivation)

It has not been deposited and we are still working on the genome sequencing. The protein sequence has been submitted to NCBI.

9. This production medium (3L) was inoculated with 2% (v/v) B. licheniformis and incubated at

99 60°C for 7 days. Author must mention other production parameter such as RPM and pH for protease production.

Thank you for feedback. The suggested correction has been made to the manuscript. 

10. Correct the sentence, line 102, further in next line precipitated enzyme pellets were suspended in 0.05M Tris-HCl 104 buffer, pH 7.5, and assayed for protein content and enzyme activity. The precipitated enzyme was dialyzed using vivaspin utlraspin centrifugal dialysis tubes with cut off 10 kMW to remove excessive salts. Salts interfere with enzyme activity so it is necessary to remove the excess salt from the suspended buffer before taking the enzyme activity. The sentence mentioned (line 103) indicate that, enzymatic activity was assayed first then dialysis was performed. Change the sequences of experiment and rewrite the section.

 it was written mistakenly wrong and now corrected in manuscript. 

11. Protease activity method and calculation should be explained in brief for readers

The Goose formula has been added as reference for understanding.

Line 110 Please add space between numbers and unit, for instance 280 nm (space between 280 and nm, please make further corrections in all the text.)

The spacing among units and words is corrected. 

12. Line 121 Digestion protocol not provided

The reference of digestion has been attached as link to the reference.

13. Section 2.4 what was the peptide concentration for MALDI-TOF MS/MS analysis?

The tryptic digest with the concentration of 1:1 (0.1 ug/ul) was added. This has been added in the manuscript as well. 

14. Line 130 how the database search was performed? Any specific genera or generic search kindly mention?

The peptide confirmation was obtained by MS/MS analysis of a tryptic peptide peak with m/z 1108.496. MS/MS showed that this peptide is identical to the peptide covering amino acid residue 342-351 (KHPNLSASQVRN) of the apr gene product (Mascot score 80) (Fig 1B). The Unipept analysis revealed that peptide was hitherto solely found in B. licheniformis Subtilisin (37).

15. Line 132 Mention the peptide charge used in MASCOT search

A positive charge was selected as the mentioned in the reference number 26. 

16. Kindly provide peptide sequence showing identity with protease of Bacillus licheniformis along with score obtained in MASCOT search.

The score of MASCOT is updated to be 80 and the sequence is added to the manuscript.

17. Reason for selection of clay as support material?

Already a part of poultry feed as mentioned in the Introduction omits the hassle of approving the content from the poultry feed federations 

18. Line 148 significance of HCl pretreatment?

The use of HCl remove the unnecessary charged particles from the clay to make it more susceptible for binding of protease to the support material.

19. Line 168 correct as supernatant

The corrections have been made as per suggestions thanks for highlighting it. 

20. Line 208 -211” Likewise, proteases were produced by B. licheniformis 209 PB1 (33) as well but under varying production parameters. Among them, physicochemical 210 parameters were a key determinant of varying production of protease in quantity and activity from 211 different species:. Rewrite the sentences for better clarity

The suggested corrections have been made to the reviewed document. 

21. Section 3.1 before mentioning the fold activity, author should mention the actual enzyme activity in terms of IU/ml

The actual enzyme units have been updated in the manuscript as per suggestion.

22. Section 3.1 line 207 “highest enzyme activity observed on seventh day” is the activity reduced or stable ,what was the reason for the reduction of activity explain in brief (1-2)lines.

The experiment was a batch culture so after a specific duration when all of the nutrients are drained the overall specific activity tends to drop indicative of the fact the all of the nutrients have be used.

23. Line 221“The peptide confirmation was obtained by MS/MS analysis of a tryptic peptide

peak with m/z 1108.496. MS/MS showed that this peptide is identical to the peptide covering amino acid residue 342-351 (K.HPNLSASQVR.N) of the apr gene product (Mascot score 80)” kindly justify the statement.

A reference is linked as per the information and additional information is also added as per my understanding. Any suggestions for improvements will be welcomed.

24. Section 3.3 line 235 Reason for better residual activity in bentonite immobilized subtilisin compared to other support used in the study?

 The residual activity improvement has been highlighted in the discussion section.

25. Author studied the effect of change in buffer Molarity, and found at 1M strength the activity was maximum, at what temperature this activity was measured. Include this information.

The temperature has been updated in the manuscript as 50C. 

26. Line 257, It has mentioned the enzyme is active at higher temperature range up to 70°C after 30 min of incubation. The loss in activity must be compared with the maximum activity at temperature 50 0 C.

These trends will be better discussed in a separate study of immobilization kinetics and parameter characteristic evaluation of free and adsorbed subtilisin.

27. Section 3.4.4, is not explained adequately. The adsorption of enzyme on a selected solid support should be explain properly.

The relevant information has been updated as per the requirement. 

28. Line 331“only a limited amount of activity was lost (how much).” check the statement and rewrite.

5% loss 

29. Section 3.6 I strongly suggest authors to compare the data of poultry trials with previously reported results. This may add merit to the present study and would help readers to understand the advantage of the work.

 The testing is unique as the combination of support with enzyme is not reported.

30. Section 3.5 This section should be more descriptive rather than straight forward observation of docking results. Mention about the ligand internal coordinates in grid for protein pocket calculation, score and binding energy with comparing where applicable.

31. Spell check the entire manuscript.

Checked and corrected

32. Check the units properly and follow standard units.

Checked and corrected 

33. Check your reference format. It should be uniform.

Reviewer #3: Manuscript number: PONE-D-21-14945

Title: Immobilization and docking studies of Thermophilic Carlsberg Subtilisin on Bentonite

as a poultry feed supplement

The manuscript entitled ‘Immobilization and docking studies of Thermophilic Carlsberg Subtilisin on Bentonite as a poultry feed supplement' is an interesting piece of work that gives details about immobilized thermophilic Carlsberg Subtilisin on bentonite and its application to poultry feed supplement'. They prepared the Carlsberg Subtilisin enzyme, and then immobilized on to bentonite surface by physical adsorption, and then feed to poultry feeding. The results are deserved to be published in enzyme technology area. However, the level of results and discussion is not considered enough to merit its publishing in PLOS ONE. Some of the suggested changes to be made are given below.

Major

Q1. The authors should describe advantage of bentonite relevant with enzyme immobilization. This discussion would be very important to address originality of the manuscript.

The discussion section includes all the necessary information highlighting the immobilization of subtilisin to bentonite and its impact on the poultry in terms of weight gain. 

Q2. Physico-chemical characterization of bentonite should be discussed.

A separate study is designed to discuss these characteristics, Raman, FTIR, XRD was performed with and without the protein immobilizations.

Q3. The author conclude that bentonite is better than carbon materials for protease immobilization. However, depending on experimental condition, the result might be changed. This information must be clear in the manuscript.

The carbon based diet formulation of proteases has a drawback of fungal contamination increasing the risks to birds.

Q4. Provide the supporting literature information about bentonite for enzyme immobilization (additional comparison Table) to justify the significance of this study.

The Bentonite is mostly used in feed as a wet liter controlling agent, making this easier to choose and found it to be a good immobilizing agent for subtilisin.

Q5. Was determined the full loading of immobilized enzyme prepared under the optimized immobilization conditions? This information must be clear in the manuscript.

The optimized conditions were observed to make the loading for immobilization and is addressed in the manuscript as well. I have added the free enzyme and immobilized enzyme comparison to highlight the impact of immobilized subtilisin to the bentonite. 

Q6. The loading capacity of enzyme immobilizaed prepared and the amount of attached enzyme: What were the optimum conditions? Please use numbers, no qualitative statements (high, good, higher, etc).

The suggested comments have been addressed in the reviewed document and suggestion changes have been made. 

Q7. The kinetics and specific activity of immobilized protease on the bentonite based on the enzyme loading need to be more quantitative and should be given and discussed in the manuscript: Authors need to compare these results with other results in the literature.

Indeed, a point to ponder, these results will be better elaborated in another study designed to evaluate the comparative analysis of free and adsorbed subtilisin separately. 

Q8. Comparison of the free and immobilized enzyme in terms of kinetic parameters: Authors need to compare these results with other results in the literature.

These parameters were studied and designed for another manuscript and shall be published with elaborated discussion of kinetic parameters and comparisons. 

Q9. The activity and stability of immobilized enzymes in bentonite carrier: The new biocatalysts presented were compared with a commercial material?? This information must be clear in the introduction.

The suggested information has been added to the introduction sections as well. 

Q10. The influence of metal ions and some inhibitors on the enzyme activity were examined?

The effects were studied in another studied to be published in another paper. 

Q11. The effect of the detergents and organic solvents was studied?

The effect of detergent and organic solvents were tested in another study

Q12. All discussion sections relevant with enzyme immobilization are very week, authors can improve it by suggesting the above citations such as how various in loading? why high activity? kinetic changes? And low reusability (may be leaching), what benefits of this system? etc.

The suggested changes have been made to the document.

Q13. Bentonite is mineral materials which might affect weight gain to chicken. However, the author didn’t include any proper experimental condition to discuss with potential synergetic effect of them. This information must be clear in the manuscript.

The amount of bentonite was balanced out in the feed controls.

---

## [Decision Letter · Decision Letter 1]

3 Feb 2022

PONE-D-21-14945R1Immobilization and docking studies of Carlsberg subtilisin for application in poultry industryPLOS ONE

Dear Dr. ali,

Thank you for submitting your manuscript to PLOS ONE. After careful consideration, we feel that it has merit but does not fully meet PLOS ONE’s publication criteria as it currently stands. Therefore, we invite you to submit a revised version of the manuscript that addresses the points raised during the review process.

We look forward to receiving your revised manuscript.

Kind regards,

Sanjay Kumar Singh Patel, Ph.D.

Academic Editor

PLOS ONE

Additional Editor Comments:

The authors should satisfy the Reviewer 3.

Please address his comments as response carefully as per PONE-D-21-14945 (previous submission). 

Reviewers' comments:

Reviewer's Responses to Questions

**Comments to the Author**

1. If the authors have adequately addressed your comments raised in a previous round of review and you feel that this manuscript is now acceptable for publication, you may indicate that here to bypass the “Comments to the Author” section, enter your conflict of interest statement in the “Confidential to Editor” section, and submit your "Accept" recommendation.

Reviewer #1: All comments have been addressed

Reviewer #2: (No Response)

Reviewer #3: (No Response)

2. Is the manuscript technically sound, and do the data support the conclusions?

Reviewer #1: Yes

Reviewer #2: Yes

Reviewer #3: Partly

3. Has the statistical analysis been performed appropriately and rigorously? 

Reviewer #1: N/A

Reviewer #2: Yes

Reviewer #3: Yes

4. Have the authors made all data underlying the findings in their manuscript fully available?

Reviewer #1: Yes

Reviewer #2: Yes

Reviewer #3: No

5. Is the manuscript presented in an intelligible fashion and written in standard English?

Reviewer #1: Yes

Reviewer #2: Yes

Reviewer #3: Yes

6. Review Comments to the Author

Reviewer #1: Author has revised the manuscript according to comments. Manuscript is recommended for publication as it is.

Reviewer #2: The quality of the manuscript has improved as a result of these revisions. I believe the manuscript (PONE-D-21-14945) entitled "Immobilization and docking studies of Carlsberg subtilisin for application in poultry industry" is now acceptable for publication in the PLOS ONE journal.

The following are the minor comments:

Comment 1: Line 32 abstract section: "The highest free enzyme activity () was observed." Correct the sentence.

Comments 2: Follow the guidelines in the writing units, and check the spacing throughout the manuscript.

Comment 3: For a clearer presentation and a more polished paper, capitalization, abbreviation, spelling, and spacing should be revised. The author should check the manuscript carefully.

Reviewer #3: The revised manuscript's corrections are not fully improved. Therefore, I recommend the article can be rejected.

---

## [Author Response · Author response to Decision Letter 1]

18 May 2022

PONE-D-21-14945

Immobilization and docking studies of Thermophilic Carlsberg Subtilisin on Bentonite as a poultry feed supplement

PLOS ONE

Reviewer 2:

Thank you for careful evaluation of the Manuscript. I have tried my best to make all the suggested changes to the manuscript.

Reviewer #3: Manuscript number: PONE-D-21-14945

Title: Immobilization and docking studies of Thermophilic Carlsberg Subtilisin on Bentonite

as a poultry feed supplement

The manuscript entitled ‘Immobilization and docking studies of Thermophilic Carlsberg Subtilisin on Bentonite as a poultry feed supplement' is an interesting piece of work that gives details about immobilized thermophilic Carlsberg Subtilisin on bentonite and its application to poultry feed supplement'. They prepared the Carlsberg Subtilisin enzyme, and then immobilized on to bentonite surface by physical adsorption, and then feed to poultry feeding. The results are deserved to be published in enzyme technology area. However, the level of results and discussion is not considered enough to merit its publishing in PLOS ONE. Some of the suggested changes to be made are given below.

Major

Q1. The authors should describe advantage of bentonite relevant with enzyme immobilization. This discussion would be very important to address originality of the manuscript.

The discussion section includes all the necessary information highlighting the immobilization of subtilisin to bentonite and its impact on the poultry in terms of weight gain. As bentonite is an inexpensive matrix that does not have toxicity and chemical reactivity towards poultry making it most suitable solid support material for immobilization of Carlsberg subtilisin enzyme for potential use in poultry industry. Moreover, Bentonite is already an approved ingredient used in Asian poultry industry with an intention to control wet droppings making it most ideal solid support. 

Q2. Physico-chemical characterization of bentonite should be discussed.

A separate comparative study is designed which addresses the Kinetics of immobilization in comparison with free enzyme. Additionally, the immobilization was confirmed by performing the physico-chemical evaluation through Raman, FTIR, SEM and XRD. The study includes the MALDI-TOF MS/MS Analysis of the Carlsberg subtilisin with O18 tagging of immobilized and free enzyme further digested with trypsin to evaluate the precise amino acid involved in binding of the protein to the bentonite. A comparative study design to evaluate the amino acid attachment pattern. The study also includes the comparative analysis of metal ions and inhibitors effect on immobilized and free enzyme.

Q3. The author conclude that bentonite is better than carbon materials for protease immobilization. However, depending on experimental condition, the result might be changed. This information must be clear in the manuscript.

The carbon-based adsorption diet formulation of proteases has a drawback of fungal contamination increasing the risks to birds. As the temperate and humid regions have greater chances of fungal growth. During experimentation upon storage for more than two weeks, we observed mold growing on the product requiring the need to add some preservatives to the formulations. Whereas this study is designed in a way that no foreign or unapproved ingredient added to feed might increase the risks to the birds. 

Q4. Provide the supporting literature information about bentonite for enzyme immobilization (additional comparison Table) to justify the significance of this study.

The Bentonite is mostly used in feed as a wet liter controlling agent, making this easier to choose and found it to be a good immobilizing agent for subtilisin. 

Q5. Was determined the full loading of immobilized enzyme prepared under the optimized immobilization conditions? This information must be clear in the manuscript.

The optimized conditions were observed to make the loading for immobilization and is addressed in the manuscript as well. I have added the free enzyme and immobilized enzyme comparison to highlight the impact of immobilized subtilisin to the bentonite but the results are already a part of another study as mentioned earlier and writing again will get plagiarized. 

Q6. The loading capacity of enzyme immobilizaed prepared and the amount of attached enzyme: What were the optimum conditions? Please use numbers, no qualitative statements (high, good, higher, etc).

The suggested comments have been addressed in the reviewed document and suggestion changes have been made. The result and discussion section contains enzyme units in U/mg. 

Q7. The kinetics and specific activity of immobilized protease on the bentonite based on the enzyme loading need to be more quantitative and should be given and discussed in the manuscript: Authors need to compare these results with other results in the literature.

The results of immobilization comparison and discussion is already a part of submitted paper as mentioned in Q4’s response. A full fledge study was designed stating the immobilization of enzyme to bentonite compared with free enzyme and discussed with respect to reported literature. 

Q8. Comparison of the free and immobilized enzyme in terms of kinetic parameters: Authors need to compare these results with other results in the literature.

These parameters were studied and designed for another manuscript and shall be published with elaborated discussion of kinetic parameters and comparisons for the same study addressed in Q4. 

Q9. The activity and stability of immobilized enzymes in bentonite carrier: The new biocatalysts presented were compared with a commercial material?? This information must be clear in the introduction.

The suggested information has been added to the introduction sections as well. The name of commercial enzyme supplement is given in methodology section referring to line number 229.

Q10. The influence of metal ions and some inhibitors on the enzyme activity were examined?

The effects were studied in another studied to be published in another paper as mentioned in Q4.

Q11. The effect of the detergents and organic solvents was studied?

The effect of detergent and organic solvents was tested in another study mentioned in Q4.

Q12. All discussion sections relevant with enzyme immobilization are very week, authors can improve it by suggesting the above citations such as how various in loading? why high activity? kinetic changes? And low reusability (may be leaching), what benefits of this system? etc.

The suggested changes have been made to the document. Majorly All of the concerning questions are related to another study that I have already designed for the second paper mentioned in Q4. 

Q13. Bentonite is mineral materials which might affect weight gain to chicken. However, the author didn’t include any proper experimental condition to discuss with potential synergetic effect of them. This information must be clear in the manuscript.

The amount of bentonite was balanced out in the feed controls. A negative and positive controls were run in parallel to address the effect. Also, Bentonite is already being used in poultry feed to reduce the wet liter control. We added same amount of unabsorbed bentonite to evaluate the difference between both. We added same amount of bentonite to the commercial enzyme supplement test to reduce the false positive and negative results. The gist is highlighted in the manuscript line number 230-232.

---

## [Decision Letter · Decision Letter 2]

27 May 2022

Immobilization and docking studies of Carlsberg subtilisin for application in poultry industry

PONE-D-21-14945R2

Dear Dr. ali,

We’re pleased to inform you that your manuscript has been judged scientifically suitable for publication and will be formally accepted for publication once it meets all outstanding technical requirements.

Kind regards,

Afsheen Aman, Ph.D.

Academic Editor

PLOS ONE

Additional Editor Comments (optional):

Reviewers' comments:

Reviewer's Responses to Questions

**Comments to the Author**

1. If the authors have adequately addressed your comments raised in a previous round of review and you feel that this manuscript is now acceptable for publication, you may indicate that here to bypass the “Comments to the Author” section, enter your conflict of interest statement in the “Confidential to Editor” section, and submit your "Accept" recommendation.

Reviewer #3: All comments have been addressed

2. Is the manuscript technically sound, and do the data support the conclusions?

Reviewer #3: Partly

3. Has the statistical analysis been performed appropriately and rigorously? 

Reviewer #3: Yes

4. Have the authors made all data underlying the findings in their manuscript fully available?

Reviewer #3: Yes

5. Is the manuscript presented in an intelligible fashion and written in standard English?

Reviewer #3: Yes

6. Review Comments to the Author

Reviewer #3: I suggest it can be accepted for publication.

The revised manuscript's corrections are improved. The justification and discussion are clearer. Therefore, I recommend the article can be accepted.

7. PLOS authors have the option to publish the peer review history of their article (what does this mean?). If published, this will include your full peer review and any attached files.

Reviewer #3: No

---

## [Editor Report · Acceptance letter]

11 Apr 2023

PONE-D-21-14945R2 

Immobilization and docking studies of Carlsberg subtilisin for application in poultry industry 

Dear Dr. Ali:

I'm pleased to inform you that your manuscript has been deemed suitable for publication in PLOS ONE. Congratulations! Your manuscript is now with our production department. 

Kind regards, 

on behalf of

Dr. Afsheen Aman 

Academic Editor

PLOS ONE